# Twist-angle dependence of moiré excitons in WS$_2$/MoSe$_2$ heterobilayers

Long Zhang [1], Zhe Zhang[1,2], Fengcheng Wu [3✉], Danqing Wang[1], Rahul Gogna[4], Shaocong Hou[5], Kenji Watanabe [6], Takashi Taniguchi [7], Krishnamurthy Kulkarni [5], Thomas Kuo[1], Stephen R. Forrest [1,5] & Hui Deng [1,4✉]

Moiré lattices formed in twisted van der Waals bilayers provide a unique, tunable platform to realize coupled electron or exciton lattices unavailable before. While twist angle between the bilayer has been shown to be a critical parameter in engineering the moiré potential and enabling novel phenomena in electronic moiré systems, a systematic experimental study as a function of twist angle is still missing. Here we show that not only are moiré excitons robust in bilayers of even large twist angles, but also properties of the moiré excitons are dependant on, and controllable by, the moiré reciprocal lattice period via twist-angle tuning. From the twist-angle dependence, we furthermore obtain the effective mass of the interlayer excitons and the electron inter-layer tunneling strength, which are difficult to measure experimentally otherwise. These findings pave the way for understanding and engineering rich moiré-lattice induced phenomena in angle-twisted semiconductor van der Waals heterostructures.

[1] Physics Department, University of Michigan, 450 Church Street, Ann Arbor, MI 48109-2122, USA. [2] State Key Laboratory of Surface Physics, Department of Physics, Fudan University, 200433 Shanghai, China. [3] Condensed Matter Theory Center and Joint Quantum Institute, Department of Physics, University of Maryland, College Park, MD 20742, USA. [4] Applied Physics Program, University of Michigan, 450 Church Street, Ann Arbor, MI 48109-1040, USA. [5] Department of Electrical Engineering and Computer Science, University of Michigan, 450 Church Street, Ann Arbor, MI 48109-1040, USA. [6] Research Center for Functional Materials, National Institute for Materials Science, 1-1 Namiki, Tsukuba 305-0044, Japan. [7] International Center for Materials Nanoarchitectonics, National Institute for Materials Science, 1-1 Namiki, Tsukuba 305-0044, Japan. ✉email: wufcheng@umd.edu; dengh@umich.edu

A tomically thin heterostructures created by stacking van der Waals materials mark a new frontier in condensed matter physics[1–3]. When two monolayer crystals of the same lattice symmetries overlay on each other, a moiré superlattice may form due to a small mismatch in their lattice constants or angular alignment[4,5]. The latter—the twist angle between the two layers—provides a powerful tuning knob of the electronic properties of the heterostructure. Seminal results have been obtained in twisted bilayer graphene, where superconducting and correlated insulating states are created by fine control of the twist angle[6–9]. In semiconductors, such as transition metal dichalcogenides (TMDC) heterobilayers, the moiré lattice has a period on the length scale of an exciton, thereby providing a unique opportunity to create coupled exciton lattices hitherto unavailable in any other systems. A wide variety of phenomena, tunable with the twist angle, may become possible, ranging from single quantum-dot arrays and topological bands to strongly correlated states[10–14].

To search for the effects of moiré lattices on excitons, split-exciton states have been reported in TMDC bilayers with very small twist angles, demonstrating localization of exciton states likely in moiré supercells[15–18]. However, increasing the twist angle has led to the suppression of measurable features of moiré excitons. In WS$_2$/MoSe$_2$ heterobilayers, it was suggested that the resonant interlayer hybridization amplifies the moiré superlattice effects on the electronic structure[19]; yet only a single resonance was resolved as the twist angle deviates significantly from 0° or 60°[18]. Existence of moiré superlattice for exciton in large-twist-angle bilayers and nontrivial effects of the twist-angle on excitons remain largely unexplored in experiments.

In this work, we show moiré excitons in heterobilayer of a wide range of twist angles and demonstrate tuning of their properties by the moiré lattice or the twist angle. Utilizing the inter- and intralayer hybrid excitons in WS$_2$/MoSe$_2$ bilayers, we reveal the formation of moiré reciprocal lattices with Brillouin zones of different sizes at different twist angles. We furthermore show how the moiré reciprocal lattices drastically change the properties of the moiré excitons, such as their resonance energies, oscillator strengths, and inter-/intralayer mixing. The twist-angle dependence of the moiré exciton states is well-explained by an analytical theory model based on band folding in the moiré lattice, which also consistently explain the dependence on the spin–orbit splitting of the conduction band, valley selection rules, atomic stacking orders, and the lattice symmetries. Comparing the experimental results with the model, we obtain the effective mass of the interlayer excitons, the interlayer electron-tunneling strength.

## Results

The devices used in this work are WS$_2$/MoSe$_2$ heterobilayers with different twist angles $\theta$, capped by few-layer hexagonal boron nitride (hBN). Details of sample fabrication and calibration of $\theta$ have been described elsewhere[20,21] and provided in the "Method" section. Figure 1a shows the optical microscope image of a heterobilayer, where the sharp edges of two monolayers are aligned. The twist angle is $\theta = 59.8° \pm 0.3°$, determined optically using polarization-dependent second-harmonic-generation measurements[20,21] (see Supplementary Figs. 1 and 2 and Supplementary Notes 3 and 4 for details).

**Identification and analysis of inter- and intralayer hybrid excitons**. We first characterize exciton hybridization in closely aligned heterobilayers, with a twist angle $\theta \sim 0°$ or 60°. In such bilayers, the Brillouin zones of the two layers closely overlap in momentum space to form nearly direct bandgaps for both the inter- and intralayer transitions (top panels of Fig. 1b). At the

same time, the hole band offset is large, but the conduction-band offset is small between WS$_2$ and MoSe$_2$ (middle panels of Fig. 1b). Therefore interlayer electron tunneling is expected between states of the same spin and valley, which leads to hybridization between the corresponding intra- and interlayer exciton transitions that share the same hole state (bottom panels of Fig. 1b).

Making use of the large difference in oscillator strength between spatially direct and indirect excitons, we identify the formation of hybrid states via the reflectance contrast (RC) spectra at 4 K: $RC = \frac{R_{sample} - R_{sub}}{R_{sub}}$, where $R_{sample}$ and $R_{sub}$ are reflection spectrum taken from sample and substrate respectively (see Supplementary Fig. 3 and Note 5). The interlayer exciton has an oscillator strength two to three orders of magnitude weaker than that of the intralayer exciton, due to separation of the electron and hole wavefunction[22–24], so it is typically too weak to be measurable in absorption or RC spectroscopy where the noise level is typically 1% or higher (Supplementary Fig. 4 and Note 6). However, when interlayer excitons hybridize with intralayer ones via electron or hole tunneling, the hybrid states acquire an oscillator strength through the intralayer exciton component. Therefore, we can identify the hybrid excitons via their spectral weight in the absorption spectra of the heterobilayer.

As shown in Fig. 1c, the MoSe$_2$ monolayer region of the device (as marked on Fig. 1a) shows a strong intralayer MoSe$_2$ A exciton resonance near 1.65 eV, while the WS$_2$ monolayer has no exciton resonances nearby. In the bilayer, stacking of the WS$_2$ layer is expected to lead to a red shift of MoSe$_2$ A exciton resonance[18] while also introducing an interlayer exciton transition, between an electron in WS$_2$ and a hole in MoSe$_2$. The interlayer exciton has a negligible oscillator strength and should not be observable in RC. However, two clearly resolved resonances appear in our bilayer, both with significant spectral weight (top two spectra in Fig. 1c). The same two resonances are also measured in photoluminescence (see Supplementary Fig. 5 and Note 7). We therefore identify them as the inter- and intralayer hybrid states, the lower (LHX) and upper hybrid excitons (UHX). Both LHX and UHX inherit an oscillator strength from their intralayer component[18], with the ratio $f_{LHX}/f_{UHX}$ controlled by their intralayer exciton fractions, which in turn is controlled by the energy detuning $\delta = E_{IX} - E_X$ between the uncoupled interlayer ($E_{IX}$) and intralayer ($E_X$) resonances. Therefore $f_{LHX}/f_{UHX}$ greater or less than one corresponds to positive or negative detuning $\delta$. There are multiple pairs of intra- and interlayer excitons that can hybridize. We focus on the transition region of MoSe$_2$ A exciton first and label these states as MoA excitons, of which the hole is always in the highest MoSe$_2$ valence band. Other pairs will be analyzed later.

As clearly seen in Fig. 1c, in the R-stacking bilayers ($\theta = 2.1°$), $f_{LHX}/f_{UHX} > 1$, suggesting the uncoupled interlayer state lies above the intralayer one, or $\delta_R > 0$. In contrast, in the H-stacking bilayer ($\theta = 59.8°$), $f_{LHX}/f_{UHX} < 1$, suggesting $\delta_H < 0$. These results are consistent with the spin selection rules of the excitonic transitions illustrated in Fig. 1b[25]. Assuming the interlayer exciton-binding energy is about the same in R- and H-stacking bilayers[26], and the difference $\delta_R - \delta_H$ is comparable to the spin–orbit splitting of WS$_2$ conduction band (Fig. 1b).

To analyze the results quantitatively, we first obtain the energies, $E_{LHX}$ and $E_{UHX}$, and oscillator strengths of the hybrid states by fitting the RC spectra using the transfer matrix method, where the hybrid excitons are modeled as Lorentz oscillators (see "Methods", Supplementary Fig. 3 and Note 5)[25,27]. The fitted spectrum agrees well with the data, as shown in Fig. 1c. Describing the hybrid modes with the coupled oscillator model, we have $E_{LHX} - E_{UHX} = -\sqrt{4J^2 + \delta^2}$, and $\frac{f_{LHX}}{f_{UHX}} = \frac{\sqrt{\delta^2 + 4J^2} + \delta}{\sqrt{\delta^2 + 4J^2} - \delta}$ (see

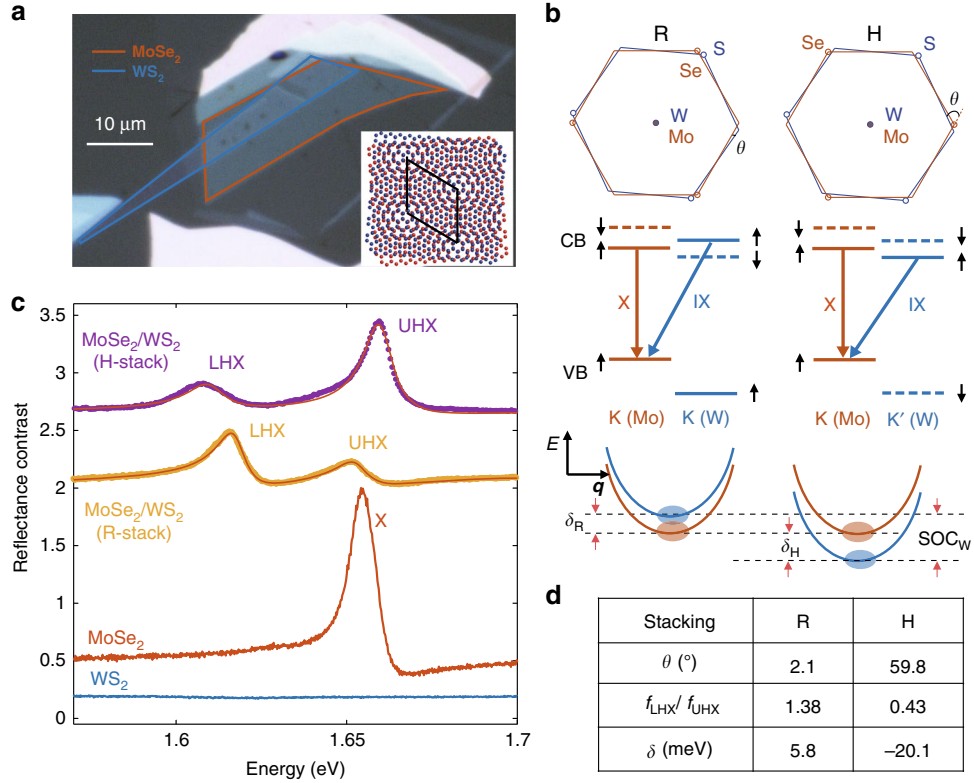

**Fig. 1 Hybrid excitons in rotationally aligned WS₂/MoSe₂ bilayers. a** An optical microscope image of a hexagonal boron nitride (hBN)-capped heterobilayer. Red and blue solid lines outline the MoSe₂ and WS₂ monolayers, respectively. The inset is an atomistic model of the WS₂/MoSe₂ heterobilayer. Red and blue spheres represent MoSe₂ and WS₂ monolayers, respectively. The black solid diamond represents the moiré unite cell. **b** Top panels illustrate a unit cell of R-stacking (left) and H-stacking (right) WS₂/MoSe₂ bilayers. Middle panels depict the corresponding band alignment of WS₂ and MoSe₂, where X labels the intralayer transition and IX labels the nearly resonant interlayer transition that shares the same hole state. Solid and dashed lines correspond to states of opposite spins. Bottom panels illustrate the alignment between an intralayer MoSe₂ A exciton state (red) and the interlayer exciton state (blue) that it hybridizes with. **c** Reflectance contrast (RC) spectra for, from bottom to top, a monolayer WS₂ (blue), monolayer MoSe₂ (red), R-stacking bilayer (orange), and H-stacking bilayer (purple). The dots are the data, and solid lines are fits. The spectra are displaced vertically for easier reading. **d** Summary of the fitted ratio of the oscillator strength between LHX and UHX, and the corresponding detuning $\delta$.

Supplementary Note 1 for details). Thereby using the fitted $E_{\text{LHX,UHX}}$ and $f_{\text{LHX,UHX}}$, we can obtain $\delta$ and $J$. As summarized in Fig. 1d, we obtain $J \sim 20$ meV for both R- and H-stacking and $\delta_R - \delta_H = 25.9 \pm 0.5$ meV, consistent with the spin–orbit splitting of WS₂[28], confirming the hybrid states are formed by spin-conserved interlayer electron tunneling.

**Twist-angle dependence of moiré-lattice-induced hybrid excitons.** To study tuning of the hybrid excitons by the moiré lattice, we perform the same measurements and analysis as discussed above on 30 samples with different twist angles, and obtain how the exciton energies, oscillator strengths and interlayer tunneling vary with the changing moiré lattice. We define $\theta_0 < 30°$ as the angular deviation from aligned bilayers of R- or H-stacking. $\theta_0 = |\theta|$ for R stacking and $\theta_0 = |60° - \theta|$ for H-stacking.

As shown in Fig. 2a, the MoA hybrid exciton doublets are clearly resolved for $\theta_0$ up to 6°, which would correspond to a tuning of the moiré lattice constant by nearly threefold[29]. The spectral weights of the doublets evolve continuously with the twist angle, reflecting the continuous increase of $f_{\text{LHX}}/f_{\text{UHX}}$ and $\delta$ with $\theta_0$ (middle panel of Fig. 2c). At the same time, the interlayer coupling $J$ decreases continuously (bottom panel of Fig. 2c). These observations show clearly moiré lattice induce hybridization and tuning of intralayer and interlayer excitons, as we explain below.

We illustrate in Fig. 2b the MoA exciton bands at different twist angles, corresponding to the six samples shown in Fig. 2a. The intralayer MoSe₂A exciton transition (red band) remains direct, with the band minimum at zero center-of-mass momentum $q_X \sim 0$, irrespective of the twist angle. It is close in energy with the interlayer exciton formed by a hole from the same MoSe₂ valence band but an electron from a WS₂ conduction band. This interlayer exciton band has the band minimum also at zero center-of-mass momentum: $q_{IX} \sim 0$, when $\theta \sim 0°$ ($\theta_1$ in Fig. 2a, b) or 60° ($\theta_6$ in Fig. 2a, b), neglecting the small lattice constant mismatch.

As the two lattices rotate relative to each other by $\theta$ ($\theta_2$ to $\theta_5$ in Fig. 2a, b), the Brillouin zones of MoSe₂ and WS₂ also rotate by $\theta$. The interlayer exciton band minimum shifts away from the intralayer exciton band minimum by momentum $K_W - K_M$ for R stacking, where $K_M$ and $K_W$ are, respectively, the Brillouin zone corners for MoSe₂ and WS₂ layers. Due to this momentum mismatch, hybridization between intralayer MoSe₂A excitons and the interlayer state at the band minimum is not allowed.

However, interlayer electron tunneling in the moiré lattice can lead to the formation of new moiré miniband states to hybridize with the optically bright intralayer excitons. As illustrated in Figs. 2b and 3c, three interlayer excitons $|q_i\rangle_{IX}$ overlap with the optically bright intralayer exciton, where the center-of-mass momentum $q_i$, measured relative to the band minimum of interlayer exciton, correspond to $q_1 = K_M - K_W$ for R stacking,

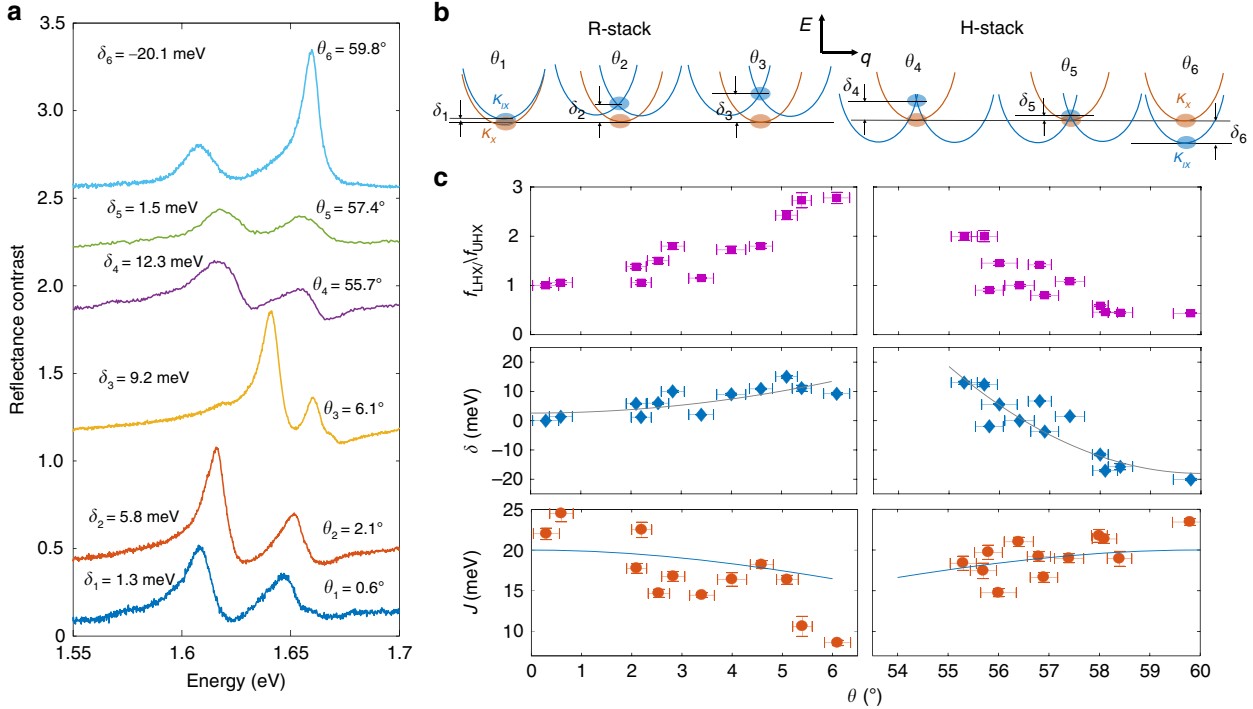

**Fig. 2 Twist-angle dependence of the hybrid excitons. a** RC spectra of bilayers of different twist angles $\theta_i$, for $i = 1$–6. The corresponding $\theta_i$ and extracted detuning $\delta_i = E_{IX,i} - E_{X,i}$ are labeled by each spectrum. The spectra are displaced vertically for easier reading. **b** Schematics of MoA intralayer (red) and interlayer (blue) exciton bands at the different twist angles $\theta_i$. The interlayer exciton band is displaced in the momentum space with increasing $\theta_i$. The moiré superlattice leads to band folding and formation of a new interlayer exciton state at the $\Gamma$ point $q = 0$ (blue oval), with the same angular momentum as the intralayer exciton state (red oval). **c** Ratio of the oscillator strengths of LHX$_{MoA}$ and UHX$_{MoA}$, detuning, and inter- and intralayer exciton coupling strength as a function of the twist angle $\theta$, obtained from the RC spectra. The gray solid lines in the middle panel are quadratic fits based on Eq. (1). The blue solid lines in the bottom panel are the theoretical values based on Eq. (2).

with $q_{2,3}$ connected to $q_1$ by $2\pi/3$ and $4\pi/3$ rotations, respectively, via moiré reciprocal lattice vectors. These three interlayer states are offset from their band minimum by the kinetic energy $\hbar^2 q_i^2/(2M_{IX})$, for $i = 1, 2, 3$, and $M_{IX}$ the total mass of interlayer exciton. These three states can couple due to the moiré lattice and therefore, superpose to form moiré miniband states, of which one interlayer exciton state shares the same angular momentum as the intralayer MoSe$_2$A exciton at $q_X \sim 0$, giving rise to the hybrid doublet we observe in angularly misaligned bilayers (see Supplementary Note 2).

When $\theta$ deviates more from 0° or 60°, the interlayer exciton formed in the moiré lattice continuously blueshifts because of the increasing kinetic energy, which explains the measured continuous blueshift of the LHX and UHX resonances, and the continuous increase of the spectral weight of LHX compared to UHX.

**Theoretical analysis of moiré-lattice-induced hybrid excitons.** To analyze our results more quantitatively, we develop an analytical microscopic theory based on the above understanding (see Supplementary Note 2 for details). Comparing it with the measured twist-angle dependence of the hybrid states, we obtain the key band parameters of the bilayer, including the interlayer exciton effective mass and interlayer coupling strength.

We first compare the measured detuning $\delta$ with $\theta_0$ and the interlayer exciton kinetic energy. As discussed above, $\delta$ is given by:

$$\delta(\theta_0) = \delta_0 + \frac{\hbar^2 q_1^2}{2M_{IX}}, \quad (1)$$

where $\delta_0$ is the detuning at $\theta = 0°$ or 60° for bilayers close to R- and H-stacking, respectively. $q_1$ is equal to $4\pi/(3a_M)$, and $a_M$ is the

moiré period approximated by $a_0/\sqrt{\theta_0^2 + \epsilon^2}$, for $a_0$ the monolayer lattice constant and $\epsilon$ the lattice constant mismatch $|a_0 - a_0'|/a_0$ between the two layers. Equation (1) shows that $\delta$ increases quadratically with $\theta_0$. As $\theta_0$ increases from 0° to 6°, $a_M$ changes by nearly threefold, and $\delta - \delta_0$ changes by sevenfold[29]. Fitting the measured $\delta$ vs. $\theta_0$ with Eq. (1), we find the interlayer exciton total mass $M_{IX}$ to be $(6.9 \pm 3.2)m_0$ and $(1.41 \pm 0.28)m_0$ for R- and H-stacking heterobilayers, respectively, for $m_0$ the electron free mass. These values are greater than the sum of the monolayer electron and hole effective masses. One possible reason is that the electron and hole effective masses in the bilayer may have been modified due to effects such as lattice relaxation[30].

We note that strain may lead to deviation in the twist angle and detuning. To minimize the effect of strain, we use devices of high structural integrity as verified by imaging and SHG (see Supplementary Note 4 and Supplementary Fig. 2 for details). The clear trends in the twist-angle dependence shown in Fig. 2 suggest that the effect of strain is relatively small and mainly leads to additional fluctuations in the measurement results above the measurement uncertainties (indicated by the error bars).

From our microscopic theory, we can also estimate the conduction-band interlayer tunneling parameter $w$ from the coupling strength $J$ through the relation

$$J = \frac{\sqrt{3}w}{\mathcal{A}} \sum_k \phi^*_{k+\frac{m_{h,IX}}{M_{IX}}q_1} \psi_k, \quad (2)$$

where $\phi_k$ and $\psi_k$ are, respectively, the relative-motion wavefunction for interlayer and intralayer excitons with the normalization $(1/\mathcal{A})\sum_k |\psi_k|^2 = 1$ and $(1/\mathcal{A})\sum_k |\phi_k|^2 = 1$. Here, $\mathcal{A}$ is the system area, and $m_{h,IX}$ is the hole mass for the interlayer exciton.

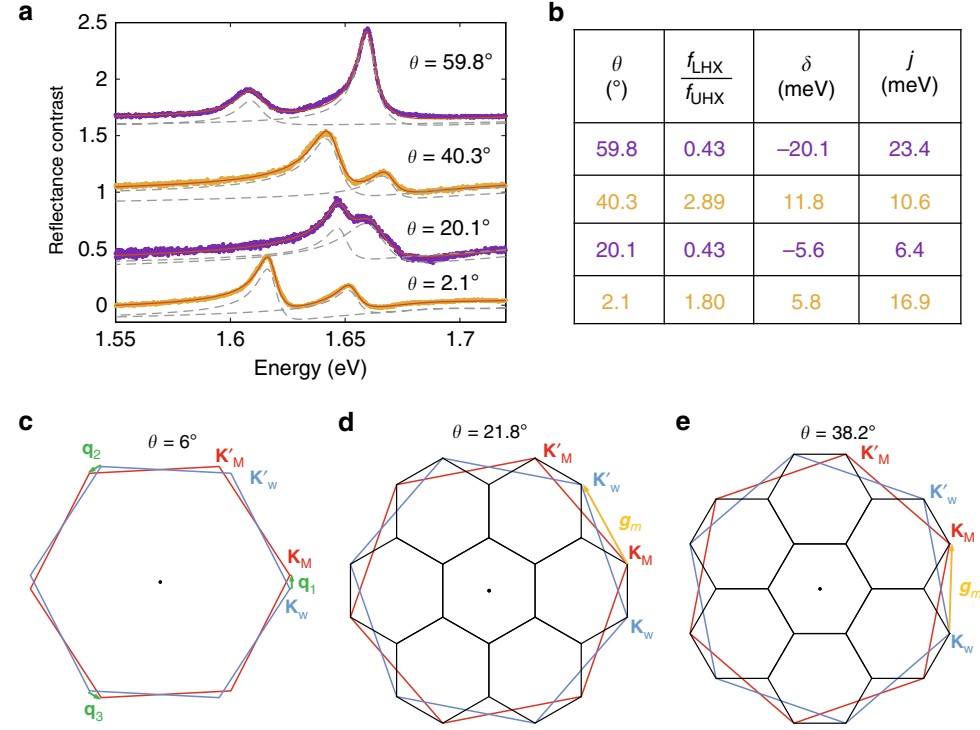

**Fig. 3 Hybridization in commensurate moiré lattices compared with aligned bilayers. a** RC spectra of bilayers with $\theta$ = 2.1°, 20.1°, 40.3°, and 59.8°. The LHX has a higher (lower) spectral weight than UHX in bilayers with $\theta$ = 2.1° and 40.3° ($\theta$ = 59.8° and 20.1°). Dots are the data, solid lines are fits, and dashed lines are the fitted individual hybrid exciton resonances. **b** Summary of the fitted parameters for the RC spectra in **a**, showing similarities between bilayers with $\theta$ = 2.1° and 40.3° and between bilayers with $\theta$ = 59.8° and 20.1°. **c–e** Schematics of the Brillouin zones of twisted bilayers. The red (blue) hexagons depict the Brillouin zones of MoSe$_2$ (WS$_2$) monolayers. In **c**, the twist angle is 6°. The green arrows indicate vectors $\boldsymbol{q}_1$, $\boldsymbol{q}_2$, and $\boldsymbol{q}_3$, which represent the momentum shift between the Brillouin zone corners of the two monolayers. In **d**, $\theta$ = 21.8°, a commensurate moiré lattice is formed, with the corresponding moiré Brillouin zone depicted by the black hexagons. The yellow arrow represents the moiré reciprocal lattice base vector that connects $\boldsymbol{K}_M$ and $\boldsymbol{K}'_W$. In **e**, $\theta$ = 38.2°, which is another commensurate angle dual to 21.8°, and $\boldsymbol{K}_M$ and $\boldsymbol{K}_W$ become equivalent states in the moiré Brillouin zone.

Because of the momentum shift $(m_{h,IX}/M_{IX})\boldsymbol{q}_1$ in the integral of Eq. (2), $J$ decreases with increasing $\theta_0$, which agrees with the experimentally observed angle dependence of $J$ (Fig. 3c). At small $\theta_0$, $J$ can be approximated by $\sqrt{3}w$. Using our experimentally measured value of $J$ at $\theta_0 \sim 0$, we estimate the interlayer tunneling $w$ to be 11.5–14.0 meV for both R- and H-stacking bilayers.

When the twist angle $\theta_0$ is greater than 6°, the hybrid exciton doublets become hard to be resolved, likely because there is a large blue detuning and the UHX has a vanishing oscillator strength (see Supplementary Fig. 6 and Note 8).

**Moiré excitons in commensurate moiré lattices at twist angles near 21.8° and 38.2°.** Remarkably, pronounced and well-resolved doublets reappear in heterobilayers with $\theta$ = 20.1° ± 0.3 and 40.3° ± 0.3, as shown in Fig. 3. In the bilayer with 20.1° twist angle, the LHX has a smaller spectral weight than UHX has, corresponding to a negative detuning ($\delta = -5.6$ meV), which is similar to H-stacking bilayers formed at $\theta \sim 60°$. In contrast, in the bilayer with 40.3° twist angle, the LHX has a larger spectral weight than UHX has, corresponding to a positive detuning ($\delta = 11.8$ meV), which is similar to R-stacking bilayers formed at $\theta \sim 0°$. In both devices, the coupling strength $J \sim 8$ meV is weaker than but of the same order of magnitude as aligned bilayers with $\theta$ close to 0° or 60°.

The revival of hybrid excitons in these two bilayers can be understood as a direct result of interlayer tunneling induced by a moiré lattice that is nearly commensurate with the monolayer lattices. The two twist angles are close to the two special commensurate angles 21.8° and 38.2°, respectively[22], where the

corresponding periodic moiré reciprocal lattices have the largest reciprocal lattice constant, $1/\sqrt{7}$ of the monolayer reciprocal lattice constant. Corners of the Brillouin zones of the two monolayers become connected by primitive moiré reciprocal lattice vectors, as illustrated in Fig. 3d and e. The MoSe$_2$ and WS$_2$ band minima overlap again in the moiré reciprocal lattice, allowing strong nearly resonant tunneling between the intra- and interlayer states. Specifically, when $\theta \approx 21.8°$, K-valley of MoSe$_2$ and K′-valley of WS$_2$ are connected by moiré reciprocal lattice vectors and are folded into equivalent momentum in the moiré Brillouin zone (Fig. 3d). The corresponding hybridized excitons have the same valley configuration as those in bilayers with $\theta \sim 60°$, which is consistent with the observed negative detuning. When $\theta \approx 38.2°$, K-valley of MoSe$_2$ and K-valley of WS$_2$ are folded into equivalent momentum in the moiré Brillouin zone (Fig. 3e), and the corresponding hybridized excitons have the same valley configuration as those in bilayers with $\theta \sim 0°$, consistent with the observed positive detuning. Moreover, since interlayer tunneling only needs one Umklapp scattering by a moiré reciprocal lattice vector, the tunneling strength remains of the same order of magnitude as in angularly aligned bilayers. Therefore, the strong revival of the hybrid excitons and their similarities with the angularly aligned bilayers show again the critical role of moiré lattice in interlayer tunneling.

**Moiré excitons formed with different intralayer excitons.** In the above discussion, we have focused on hybrid states formed with the MoSe$_2$A excitons, which feature large spectral weight, relatively narrow linewidths, and well-resolved doublets at small

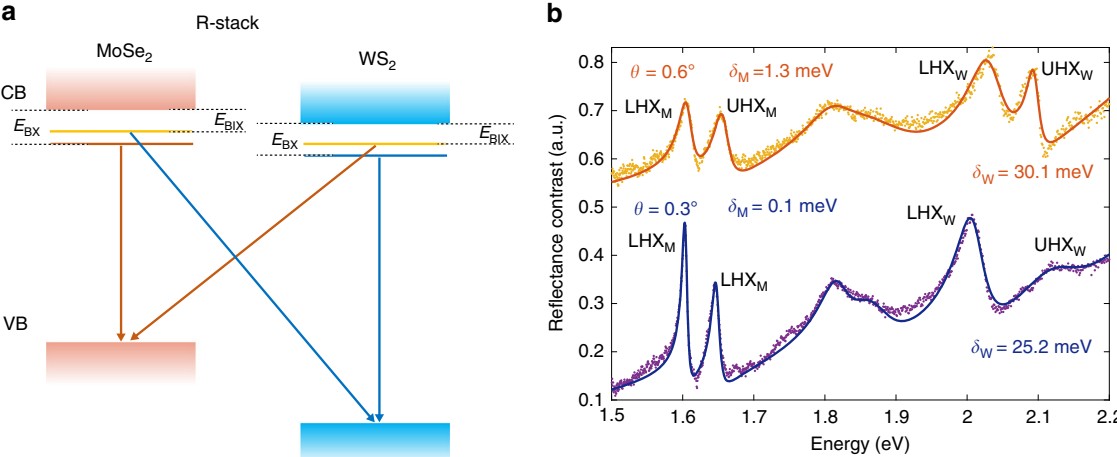

**Fig. 4 Comparison of MoA and WA hybrid states. a** Band diagram of R-stacking MoSe$_2$/WS$_2$ bilayers. The conduction and valence bands are represented by broad continuous bands. Exciton states are represented by the horizontal solid lines. Arrows represent spin-conserved exciton transitions, which are lowered in energy from the band-to-band transition by binding energy. $E_{BX}$ and $E_{BIX}$ denote the binding energies for the intra- and interlayer transitions, respectively. **b** RC spectra of both MoA and WA hybrid excitons from bilayers with $\theta \sim 0°$. Dots are the data, and solid lines are fits. The corresponding $\theta$ and detuning $\delta_M$ and $\delta_W$ obtained from fitting are labeled by each spectrum.

detunings. Hybrid states can also form with higher-energy bands, including the A excitons of WS$_2$ (WA), and B excitons of WS$_2$ and MoSe$_2$. The B excitons have broader linewidths than the A excitons; as a result, the doublets are not well resolved. The A excitons of WS$_2$ have a broader linewidth than A excitons of MoSe$_2$ and generally a larger detuning. We observe well-resolved WA doublets only in bilayers with $\theta \sim 0°$, corresponding to hybrid excitons formed by a hole in the WS$_2$ valence band and an electron tunneling between the MoSe$_2$ and WS$_2$ conduction bands (Fig. 4).

It is interesting to compare the detuning for MoA and WA states for $\theta \sim 0°$, which we label as $\delta_{MoA}$ and $\delta_{WA}$, respectively. As shown in the schematic electronic band diagram in Fig. 4a, neglecting exciton-binding energies, the detuning of the interlayer transition from the intralayer one is the same magnitude but opposite signs between the MoA and WA states. The sum of the two detuning should be zero. However, this is different from our observation that both the LHX states have larger spectral weight for both MoA and WA states. This can be understood as due to the weaker binding energy of interlayer excitons compared to intralayer ones, resulting from electron–hole separation. The difference in intra- and interlayer exciton-binding energies, $\Delta E_B^R = E_{BX} - E_{BIX}$, adds to both $\delta_{MoA}$ and $\delta_{WA}$. Assuming $\Delta E_B^R$ is approximately the same for the MoA and WA state, the sum of $\delta_{MoA}$ and $\delta_{WA}$ becomes twice of $\Delta E_B$, or, $\Delta E_B^R = 1/2(\delta_{MoA} + \delta_{WA})$. From our measurements of MoA and WA states in bilayers with $\theta < 1°$, we estimate $\Delta E_B^R$ of 10–16 meV (Fig. 4b). In comparison, values between 17 meV and 100 meV have been reported based on first principle calculations[31–34] or measurements on homo-bilayers[35–37]. The relatively small values of 10–16 meV we measured may possibly be due to difference in the materials or a smaller interlayer exciton Bohr radius as a result of strong interlayer tunneling.

## Discussion

In summary, we demonstrate hybrid states formed between momentum-direct, moiré-induced interlayer, and intralayer excitons in twisted WS$_2$/MoSe$_2$ bilayers, opening the door to studies of excitonic phenomena in twist bilayers.

Deviation of the twist angle from 0° or 60° not only does not suppress moiré excitons but provides a sensitive tuning knob of the

moiré excitons' properties. Persistence of the moiré excitons, or, the moiré lattice, is clearly manifested in the interlayer tunneling strength, which remains within the same order of magnitude over the measured range of twist angles. It is possible because momentum conservation between the twisted layers is restored by the moiré lattice, or Umklapp scattering by the moiré reciprocal lattice vector. Remarkably, while large detuning between the interlayer and intralayer states suppressed hybridization over a range of angles, pronounced hybrid moiré excitons due to strong interlayer tunneling reappears near twist angles of 21.8° and 38.2°. At these angles, moiré-lattices are formed commensurate with the mono-layer lattices, bringing angularly shifted valleys of the two mono-layers into equivalent momentum in the same moiré Brillouin zone, thereby enabling strong interlayer tunneling. The resulting hybrid exciton states resemble the features in heterobilayers with $\theta = 60°$ and 0°, respectively. These results are direct manifestations of the discrete translational symmetry of the underlying moiré super-lattice, which enables transitions that otherwise would not conserve momentum.

Since the hybrid excitons are formed in moiré reciprocal lattices, their properties dependent sensitively on the moiré period, or the twist angle. Utilizing the twist angle degree of freedom, we demonstrate tuning of the moiré exciton properties and fur-thermore obtain fundamental parameters of the bilayer system that are difficult to measure otherwise, including conduction-band splitting of WS$_2$ induced by spin–orbit coupling, the effective mass of the interlayer excitons in R- and H-stacking bilayers, interlayer electron-tunneling strength, and the difference of binding energies between intra- and interlayer excitons at the presence of interlayer tunneling.

These hybrid excitons inherit large oscillator strengths from the intralayer component that may allow strong exciton–photon coupling while, at the same time, inherit static dipole moment from the spatially indirect interlayer component that leads to long-range interactions. In a moiré lattice, both the oscillator strength and dipole interactions depend sensitively on, and can be tuned by the twist angle. The twisted WS$_2$/MoSe$_2$ bilayers may provide tunable, nonlinear, exciton and polariton lattice systems for exotic states of matter, such as topological excitons and exciton crystals, with novel applications in nanophotonics and quantum information science[10–13,38–53].

## Methods

**Sample fabrication**. Monolayer $MoSe_2$, $WS_2$, and few-layer hBN flakes are obtained by mechanical exfoliation from bulk crystals. A polyethylene terephthalate (PET) stamp was used to pick up the top hBN, $WS_2$ monolayer, $MoSe_2$ monolayer, and the bottom hBN under a microscope. After picking up all the layers, PET stamp was then stamped onto the sapphire substrate, and the PET was dissolved in dichloromethane for six hours at room temperature.

**Optical measurements**. For low-temperature measurements, the sample is kept in a 4 K cryostat (Montana Instrument). The excitation and collection are carried out with a home-built confocal microscope with an objective lens with a numerical aperture (NA) of 0.42. For reflection contrast measurement, white light from a tungsten halogen lamp is focused on the sample with a beam size of 10 μm in diameter. The spatial resolution is improved to be 2 μm by using pinhole combined with confocal lenses. The signal is detected using a Princeton Instruments spectrometer with a cooled charge-coupled camera.

## Data availability

Data are available from the authors upon reasonable request.

## Code availability

Code is available from the authors upon reasonable request.

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

## Acknowledgements

L.Z., R.G., S.H, S.F., and H.D. acknowledge support by the Army Research Office under Award W911NF-17-1-0312. H.D. also acknowledges support by the National Science Foundation under Award DMR-1838412 and by the Air Force Office of Scientific

Research under Award FA2386-18-1-4086. S.F. also acknowledges support by the U.S. Department of Energy, Office of Science, Office of Basic Energy Sciences, under Award Number DE-SC0017971. F.W. acknowledges support by Laboratory for Physical Sciences. K.W. and T.T. acknowledge support from the Elemental Strategy Initiative conducted by the MEXT, Japan, Grant Number JPMXP0112101001, JSPS KAKENHI Grant Number JP20H00354 and the CREST (JPMJCR15F3), JST.

## Author contributions

H.D. and L.Z. conceived the experiment. L.Z. and Z.Z. fabricated the device and performed the measurements. F.W. performed the modeling and calculations. L.Z. and H.D. performed data analysis. R.G. performed tunneling estimation. D.W., S.H., K.K., and T.G. assisted the fabrication. K.W. and T.T grew hBN single crystals. H.D. and S.F. supervised the projects. L.Z., F.W., and H.D. wrote the paper with inputs from other authors. All authors discussed the results, data analysis, and wrote the paper.

## Competing interests

The authors declare no competing interests.
