## [Peer Review File · Nature Communications]

REVIEWER COMMENTS

Reviewer #1 (Remarks to the Author):

The authors reported demonstration of Moire excitons in commensurate WS₂/MoSe₂ heterobilayers with large twist angles by reflection spectroscopy. The WS₂/MoSe₂ samples with different twist angles were obtained by exfoliation and stamp transfer and are all sandwiched by few-layer hBN. The precise twist angle was confirmed by polarization-dependent SHG. Reflectance spectra of all the samples were measured at 4 K. Resonances associated with the hybridized excitonic states were identified and used to probe the Moire potential formation that modulates the excitonic properties. Based on these, the authors claimed that heterobilayers with large twist angles can host Moire excitons when the two lattices are commensurate.

The results are interesting and represent a significant step forward on understanding in-plane control of photoexcitation of transition metal dichalcogenides heterobilayers - a timely topic following several breakthroughs last year. The choice of WS₂ and MoSe₂ for this purpose appears reasonable as the two materials have the furthest intralayer excitons, allowing study of hybridization of interlayer excitons with the intralayer excitons of one of the layers without complications from the other layer. The data are of high quality and the main features are well captured by the model, allowing deduction of key parameters describing the Moire excitonic states. The manuscript was well written. I support publication of this interesting work on Nature Communications if the authors can address the following comments:

1. It would be helpful for the readers if the authors could include an atomistic model of the key samples to illustrate the Moire patterns.
2. Although the trend shown in Figure 2c (middle panels) is clear, the data do not necessarily support the quadratic dependence given the large uncertainties. Could the authors comment on the potential sources of error? Can lattice strain play a role in such changes? I think it would be useful to address the strain (especially its potential impacts on the detuning).
3. Are 21.8 and 38.2 the only two large twist angles that give commensurate lattices? If so, the authors should make it clear.
4. The authors indicated sample temperature in the Method section (4 K). This should be mentioned in the main text or figure caption.

Reviewer #2 (Remarks to the Author):

This paper reports a very interesting study of hybridization of interlayer and intralayer excitons in vertically stacked hetero-bilayers as a function of twist angles. Such a hybridization is efficient due to the small conduction band offset between WS₂ and MoSe₂. Although hybridized excitons has been reported recently (in 2019 Nature on the same system, and in 2019 Science Advances for valence hybridized exciton), this paper brings two new aspects to this topic. The first is the angular dependence of hybridized excitons resulting from the exciton dispersion (from which they also determine the effective mass). The second interesting aspect is the enhanced interlayer coupling at large twist angles (21.8 or 38.2 degrees) that can lead to commensurate matching between the moire BZ and atomic BZ.

The experimental work is beautifully done and the analysis is very solid. I recommend its publication in Nature Communications after the authors answer the following questions.

- (a) One thing puzzles me is the dramatically different effective masses for interlayer excitons

being observed between the R (near 0 degree) and H (near 60 degrees) stackings ($1.4 m_0$ versus $6.9 m_0$). Since in both cases, this is related to the conduction band state hybridization, the two SO-split K-valleys should have very similar effective masses. Why would it lead to such a dramatically different effective masses in excitation bands.

(b) In the near commensurate angles, the authors show strong revival of optical transition matrix element (upto 25% of the transition probability for direct K-K alignment). On the other hand, I recall a recent Nature paper (ref 17) investigation of interlayer excitons in MoSe₂/WSe₂, at this commensurate angle, the transition probability is only 1% of the direct K-K transition. I understand the case here is for hybrid exciton. Can authors provide a quantitative analysis to explain this dramatic difference?

(c) The analysis of commensurate angles (at 21.8 and 38.2 degrees) is based on lattice matched hetero-bilayers which is applicable for MoS₂/WS₂ or MoSe₂/WSe₂. But here the lattice mismatch is 4%. Can authors justify that this analysis is equally valid?

In summary, this is an excellent paper and I recommend its publication in Nature Communications after authors address the above comments.

We thank the referees for the positive recommendation of the manuscript and the many insightful and constructive comments that have helped improving the manuscript.

We provide a point-by-point response below with the referees' comments quoted in full in blue, and revisions to the manuscript **quoted in red**.

Report of Reviewer #1---NCOMMS-17-23725-T

The authors reported demonstration of Moiré excitons in commensurate WS₂/MoSe₂ heterobilayers with large twist angles by reflection spectroscopy. The WS₂/MoSe₂ samples with different twist angles were obtained by exfoliation and stamp transfer and are all sandwiched by few-layer hBN. The precise twist angle was confirmed by polarization-dependent SHG. Reflectance spectra of all the samples were measured at 4 K. Resonances associated with the hybridized excitonic states were identified and used to probe the Moiré potential formation that modulates the excitonic properties. Based on these, the authors claimed that heterobilayers with large twist angles can host Moiré excitons when the two lattices are commensurate.

The results are interesting and represent a significant step forward on understanding in-plane control of photoexcitation of transition metal dichalcogenides heterobilayers - a timely topic following several breakthroughs last year. The choice of WS₂ and MoSe₂ for this purpose appears reasonable as the two materials have the furthest intralayer excitons, allowing study of hybridization of interlayer excitons with the intralayer excitons of one of the layers without complications from the other layer. The data are of high quality and the main features are well capture by the model, allowing deduction of key parameters describing the Moiré excitonic states. The manuscript was well written. I support publication of this interesting work on Nature Communications if the authors can address the following comments:

1. It would be helpful for the readers if the authors could include an atomistic model of the key samples to illustrate the Moiré patterns.

We thank the referee for the suggestions and have added the following schematic in the updated figure1.

Fig.1 Atomistic model of the MoSe₂/WS₂ heterobilayer. Red and blue spheres represent monolayer MoSe₂ and WS₂ respectively. The black solid diamond represents the moiré unite cell.

2. Although the trend shown in Figure 2c (middle panels) is clear, the data do not necessarily support the quadratic dependence given the large uncertainties. Could the authors comment on the potential sources of error? Can lattice strain play a role in such changes? I think it would be useful to address the strain (especially its potential impacts on the detuning).

We agree that strain may be the main source of error after counting for the measurement uncertainties shown by the error bars in Fig. 2c. Given the many complications strain can introduce, we have aimed to minimize the effect of strain in our study by using samples that are likely to have the least amount of strain. As discussed in Supplementary Information Sec. IV and Fig. S2, we made sure that, in our devices, each monolayer was transferred without folding or breaking and with minimal visible bubbles. We confirmed that, in such devices, the monolayers typically show minimal spatial variation of the crystal axis orientation and the heterobilayer also shows uniform SHG intensity, both of which are strong indicators of high structural integrity and small strain. The clear systematic trends in twist-angle dependence we observed also suggest strain indeed didn't play a dominate role.

Following the referee's suggestion, we added the following to the discussion of Fig. 2c (middle of page 7):

“We note that strain may lead to deviation in the twist angle and detuning. To minimize the effect of strain, we use devices of high structural integrity as verified by imaging and SHG (see Supplemental Material Sec. IV and Fig. S2 for details). The clear trends in the twist-angle dependence shown in Fig. 2 suggest that the effect of strain is relatively small and mainly leads to additional fluctuations in the measurement results above the measurement uncertainties (indicated by the error bars).”

It will be very interesting to study how to identify and quantitatively characterize different effects of strain on the moiré excitons. But that is beyond the scope of the current manuscript.

3. Are 21.8 and 38.2 the only two large twist angles that give commensurate lattices? If so, the authors should make it clear.

We thank the referee for the question. For two monolayers with the same lattice constant, commensurate lattices are formed at an infinite number of twist angles θ that satisfy $\cos \theta = (3m^2 + 3mr + r^2/2)/(3m^2 + 3mr + r^2)$, where m and r are coprime positive integers [see J. M. B. Lopes dos Santos, et al., PRB 86, 155449 (2012) for derivation]. However, 21.8° and 38.2° are special in that they lead to a commensurate structure with the smallest spatial period ($\sqrt{7} \times \sqrt{7}$) and largest moiré period, such that the K or K' valleys of the two monolayers become connected by a primitive vector of the moiré reciprocal lattice. Moiré physics can revive near 21.8° and 38.2°; see PRL 123, 186402 (2019) for theoretical modelling of moiré physics near large commensurate angles.

We updated the manuscript as follows:

“The two twist angles are close to the **two special** commensurate angles 21.8° and 38.2°, respectively, **where the corresponding periodic** moiré reciprocal lattices have **the largest reciprocal** lattice constant, $1/\sqrt{7}$ of the monolayer reciprocal lattice constant. Corners of the Brillouin zones of the two monolayers become connected by **primitive** moiré reciprocal lattice vectors, as illustrated in Figs. 3d and e.”

4. The authors indicated sample temperature in the Method section (4 K). This should be mentioned in the main text or figure caption.

We thank the referee for the suggestions and have added the temperature in the main text:

“Making use of the large difference in oscillator strength between spatially direct and indirect excitons, we identify the formation of hybrid states via the reflectance contrast (RC) spectra **at 4K.**”

Report of Reviewer #2---

This paper reports a very interesting study of hybridization of interlayer and intralayer excitons in vertically stacked hetero-bilayers as a function of twist angles. Such a hybridization is efficient due to the small conduction band offset between WS₂ and MoSe₂. Although hybridized excitons has been reported recently (in 2019 Nature on the same system, and in 2019 Science Advances for valence hybridized exciton), this paper brings two new aspects to this topic. The first is the angular dependence of hybridized excitons resulting from the exciton dispersion (from which they also determine the effective mass). The second interesting aspect is the enhanced interlayer coupling at large twist angles (21.8 or 38.2 degrees) that can lead to commensurate matching between the moiré BZ and atomic BZ.

The experimental work is beautifully done and the analysis is very solid. I recommend its publication in Nature Communications after the authors answer the following questions.

We thank the referee for the very positive evaluation of the work.

(a) One thing puzzles me is the dramatically different effective masses for interlayer excitons being observed between the R (near 0 degree) and H (near 60 degrees) stackings ($1.4 m_0$ versus $6.9 m_0$). Since in both cases, this is related to the conduction band state hybridization, the two SO-split K-valleys should have very similar effective masses. Why would it lead to such a dramatically different effective masses in exciton bands.

The referee raised an excellent point. We agree with the referee that the difference in the extracted effective masses for R and H stacking is surprising. The extracted effective mass ($1.4 m_0$) for H stacking is comparable with the expected exciton effective mass based on electron and hole masses of the two monolayers, while the number for R stacking ($6.9 m_0$) is much larger. We do not have a quantitative explanation, and one speculation is that the effective masses of the electrons and holes can be significantly modified due to, for example, the lattice relaxation effects (see for example Nam, N. N. T. & Koshino, M. Phys. Rev. B 96, 075311 (2017)), which can be quite different for R and H stacking.

We have updated the manuscript as follows:

“Fitting the measured δ vs θ_0 with Equation. (1), we find the inter-layer exciton total mass M_{IX} to be $6.9 \pm 3.2 m_0$ and $1.4 \pm 0.28 m_0$ for R- and H-stacking heterobilayers, respectively, for m_0 the electron free mass. **These values are greater than the sum of the monolayer electron and hole effective masses. One possible reason is that the electron and hole effective masses in the bilayer may have been modified due to effects such as lattice relaxation [Nam, N. N. T. & Koshino, M. graphene. Phys. Rev. B 96, 075311 (2017)].**”

(b) In the near commensurate angles, the authors show strong revival of optical transition matrix element (up to 25% of the transition probability for direct K-K alignment). On the other hand, I recall a recent Nature paper (ref 17) investigation of interlayer excitons in MoSe₂/WSe₂, at this commensurate angle, the transition probability is only 1% of the direct K-K transition. I understand the case here is for hybrid exciton. Can authors provide a quantitative analysis to explain this dramatic difference?

We first note that our observation is based on optical absorption spectrum, which provides a direct measurement of the optical transition probability. In Nature 567, 66 (2019), the transition corresponds to spatially indirect interlayer excitons, so only photoluminescence spectra can be measured, which provides an indirect indication of the transition probability. The resulting emission intensity could be affected by other effects related to inhomogeneity, carrier kinetics, energy relaxation pathways etc.

We currently don't have a quantitative theory for the transition probability at these large comment angles. In a continuum theory, the Umklapp process near 21.8° or 38.2° relies on the higher-harmonic Fourier component of the two-center hopping integral, which can be much weaker in strength compared to the Umklapp process near 0° or 60° . But the above comparison relies on the two-center approximation, which might break down for TMDs with complicated orbital structures. Our observation of the strong revival of hybrid excitons near 22° and 38° suggests the need for a detailed microscopic modelling of the system using density functional theories.

(c) The analysis of commensurate angles (at 21.8 and 38.2 degrees) is based on lattice matched hetero-bilayers which is applicable for MoS₂/WS₂ or MoSe₂/WSe₂. But here the lattice mismatch is 4%. Can authors justify that this analysis is equally valid?

The analysis of the moiré physics does not require the same lattice constants for the two layers and our theoretical calculations have included the lattice mismatch. Quantitatively, the lattice mismatch of 4% results in significant different moiré periods at small twist angles, but the difference becomes negligible (~1%) at 21.8° or 38.2°.

In summary, this is an excellent paper and I recommend its publication in Nature Communications after authors address the above comments.

We thank the referee for the sharp comments and hope we have addressed them satisfactorily.

Report of Reviewer #3---NCOMMS-17-23725-T

The manuscript by Zhang et al. reported the observation of hybridized state of intra- and inter-layer excitons in WS₂/MoSe₂ heterobilayers of various twisting angles. The observations are made possible by the fact that the conduction band edges of the WS₂ and MoSe₂ are nearly aligned, with a detuning comparable to the magnitude of interlayer electron hopping. The hybridization of the two exciton species is then observable from the energy splitting of the two hybridized branches, whose separation and relative weighting can be used to extract both the detuning and hopping magnitude. The hybridization has been observed and examined for the direct hopping at twisting angles near 0 and 60, and also the Umklapp assisted hopping near 20 and 40 degrees, from which the conduction band spin splitting, exciton mass, binding energy difference are extracted. Such information is very important for the understanding of optoelectronic properties in 2D TMDs. Moreover, hybrid exciton itself is of remarkable importance. Its large optical dipole and permanent electric dipole can be highly advantageous for exploration of exciton condensates as well as optoelectronic applications. I find the work suitable for Nature Communications. This version of the manuscript, on the other hand, need to be improved by addressing the following issues.

(1) In the abstract, "The twist-angle degree of freedom has been largely considered detrimental to the observation of moiré excitons." This is a rather vague statement. Moiré excitons are observed at various twisting angles near the 0, 60, and even 20 degrees (in Ref. 17). I believe the consensus is observation of interlayer exciton PL become difficult only when twisting angle gets significantly different from the above commensurate angles.

We thank the referee for the comment. We have changed the sentence to "While moiré excitons have been reported in heterobilayers, a systematic experimental study as a function of twist angle is still missing."

(2) The lineshape of the exciton resonances vary between Lorentz and anti-Lorentz. For example, in Fig. 1c, the H-stack bilayer features two resonances that are mostly Lorentz, while in the R-stacking one, the two resonances have obvious anti-Lorentz like shape. Can the authors provide some insights on what causes this difference?

We appreciate the referee carefully checking the lineshapes. The spectra in Fig.1c are reflection contrast spectra, which is defined as $R_C = \frac{R_{sample} - R_{sub}}{R_{sub}}$ (see Supplemental Material Fig.S3 for more

details). So the lineshape is determined by both R_{sample} and R_{sub} , which can not be perfect Lorentzian, but will vary with the R_{sub} with different thickness of hBN. In Fig.1c, the H stack bilayer is capped by a thin hBN of around 3 nm thick, and the resulting R_C spectrum is mostly Lorentz. The R stack bilayer is capped with an hBN of 8 nm thick, resulting in obvious anti-Lorentz like lineshapes. This can also be clearly seen in calculations of the reflectance contrast spectra of such a structure: a monolayer with a single resonance at 1.6 eV, described by a Lorentzian function, with hBN on the top of different thicknesses. As shown in the figure, the reflection contrast spectra from samples with thinner hBN will be more like Lorentzian lineshape.

Fig.2 Reflectance contrast spectra of hBN/TMDCs with a fixed exciton resonance at 1.6 eV, but different hBN thicknesses.

(3) By comparing the weight of the lower and upper hybridized states in R-stacking and H-stacking bilayers, one can conclude that the detuning of inter- and intra-layer excitons change sign in these two stacking cases. This is very useful information, from which the authors extract the spin splitting in conduction band. I wonder if the authors have taken into account the possible difference in the binding energy of interlayer exciton in R and H stacking, given their different interlayer distance?

This is an excellent question. It is typically expected that the interlayer exciton binding energy is about the same for different stacking orders, since the variation in interlayer separation is small compared to the Bohr radius (see for example a calculation in [Lu, N. et al. Nanoscale 9, 19131 (2017)]). But we agree that, without a more detailed calculation, detuning difference between the two stacking orders can only serve as a sanity check and a reference for future studies.

We have added a clarification in the updated manuscript (middle of page 5):

“These results are consistent with the spin selection rules of the excitonic transitions illustrated in Fig. 1b [24] and the difference $\delta_R - \delta_H$ ~~corresponds~~ **is comparable** to the spin-orbit splitting of WS_2 conduction band (Fig. 1b).”

(4) The formulations in several places are reminiscent of the theory paper by Yu et al. PRL 115, 187002 (2015) addressing the light coupling of interlayer exciton momentum states in twisted bilayer. The coupling to intralayer exciton state analysed here is essentially similar in terms of momentum conservation, including the Umklapp assisted process. Oscillator strength of interlayer exciton was also calculated for the first time in this paper. I think the work should be properly cited.

We thank the referee for pointing out this accidental omission and have added the reference to the updated manuscript:

Main text:

“The interlayer exciton has an oscillator strength two to three orders of magnitude weaker than that of the intra-layer exciton, due to separation of the electron and hole wavefunction [Yu et al. PRL 115, 187002 (2015)].”

“The revival of hybrid excitons in these two bilayers can be understood as a direct result of interlayer tunneling induced by a moiré lattice that is nearly commensurate with the monolayer lattices. The two twist angles are close to the commensurate angles 21.8° and 38.2° , respectively [Yu et al. PRL 115, 187002 (2015)].”

Supplementary material:

For interlayer excitons, we consider states with a generic finite center-of-mass momentum Q [Yu et al. PRL 115, 187002 (2015)].

(5) From fitting the twisting angle dependence, the authors have determined an inter-layer exciton total mass of $(6.9 \pm 3.2) m_0$ in R-stacking and $(1.41 \pm 0.28) m_0$ for H-stacking bilayers. This is somewhat surprising. One expects the exciton mass to be the sum of electron and hole masses, which are known to be a fraction of m_0 in these TMDs monolayers. Why the exciton mass can be so large, and what causes the big difference in R and H stacking?

We agree with the referee that the difference in the extracted effective masses for R and H stacking is surprising. The extracted effective mass ($1.4 m_0$) for H stacking is comparable with the expected exciton effective mass based on electron and hole masses of the two monolayers, while the number for R stacking ($6.9 m_0$) is much larger. We do not have a quantitative explanation, and one speculation is that the effective masses of the electrons and holes can be significantly modified due to, for example, the lattice relaxation effects (see for example Nam, N. N. T. & Koshino, M. Phys. Rev. B 96, 075311 (2017)), which can be quite different for R and H stacking.

We have updated the manuscript as follows:

“Fitting the measured δ vs θ_0 with Equation. (1), we find the inter-layer exciton total mass M_{IX} to be $6.9 \pm 3.2 m_0$ and $1.4 \pm 0.28 m_0$ for R- and H-stacking heterobilayers, respectively, for m_0

the electron free mass. These values are greater than the sum of the monolayer electron and hole effective masses. One possible reason is that the electron and hole effective masses in the bilayer may have been modified due to effects such as lattice relaxation [Nam, N. N. T. & Koshino, M. graphene. Phys. Rev. B 96, 075311 (2017)].”

(6) For the coupling strength J , there is a remark that it decreases with twisting angle, while being a constant at small twisting angle. This trend shall be more explicitly explained. What angle range is the constant being a good approximation? And how fast J decreases beyond this range?

We think the referee is referring to this comment in our theoretical analysis:

“Because of the momentum shift $\left(\frac{m_{h,lx}}{M_{lx}}\right)q_1$ in the integral of Eq. (2), J decreases with increasing θ_0 , which agrees with the experimentally observed angle dependence of J . At small θ_0 , J can be approximated by $\sqrt{3}\omega$.”

To address the referee’s question, we added the theoretical values of J as a function of θ in the updated Fig.2c of the manuscript, as reproduced below. It compares reasonably well with experimental data. As shown by the figure, J can be approximated by a constant for $\theta_0 < 1^\circ$.

(7) It is mentioned that the hybrid exciton doublets become hard to resolve when the twist angle is greater than 6 degree. At 6 degree, the UHX still has a significant oscillator strength. I wonder if the change is gradual at larger twisting (say 7, 8 degree). How does J change at this angle range?

In supplementary material Fig. S6, we showed spectra with twist angle of 10.5° and 11.3° , where the hybrid exciton doublets cannot be resolved. Unfortunately, we don't have samples with twist angles between 6° and 10° for R stacking sample. We have two samples with twist angle in between 50° and 54° for H stacking samples. The one with twist angle of 51.9° is shown in Fig. S6. We show the other sample with twist angle of 52.5° in the figure to the right, where only one resonance can be clearly resolved. From these data, we do not know how J changes between 6 and 10 degrees of twist angles.

Theoretically, J continues to decrease as a function of θ for large θ .

(8) The large splitting at near 20 and 40 degree twisting is also quite surprising, as these are the Umklapp process requiring the lattice to supply a reciprocal lattice vector. Typically the Fourier components of the two-center hopping integral decays very fast, and Umklapp process is much weaker in strength compared to the direct ones. For the 38.2° , 21.8° bilayer, the optical dipole of Umklapp emission is orders weaker compared to the 0° , 60° degree ones (c.f. estimation in Yu et al. PRL 115, 187002 (2015), and experiments of Seyler et al. Nature 567, 66 (2019)). I expect the hopping integral to be in a similar situation. Can the authors give more insight on why the Umklapp assisted tunnelling can be large.

We understand the referee's concern. In a continuum theory, the Umklapp process near 21.8° or 38.2° relies on the higher-harmonic Fourier component of the two-center hopping integral, which can be much weaker in strength compared to the Umklapp process near 0° or 60° . It is important to note that the above comparison relies on the two-center approximation, which might break down for TMDs with complicated orbital structures. Our observation of the strong revival of hybrid excitons near 22° and 38° calls for a detailed microscopic modelling of the system using density functional theory.

(9) The difference in intra- and inter-layer exciton binding energies is estimated to be 10-16 meV. This value is significantly lower than other literatures, including experiments, and the first principle calculations cited. As the authors mentioned, the calculations neglected interlayer tunneling. I wonder if this remark is implying the small value is because of the interlayer tunneling. I hope the authors can elaborate on this.

We looked more carefully into the recent literature and find there is a wider range of reported values, down to 17 meV [Gillen, R, *Phys. Rev. B* **97**, 1–7 (2018)], which is close to our measured values (10-16 meV). The difference may have resulted from material specific details or stronger interlayer tunneling, as we suspect interlayer tunneling may be correlated with a smaller interlayer exciton Bohr radius. But a theoretical calculation is beyond the scope of the manuscript.

We have updated the manuscript with the new reference as follows:

~~“The value is significantly lower than predictions based on first principle calculations when interlayer tunneling is neglected [29,30]. In comparison, values between 17 meV and 100 meV have been reported based on first principle calculations [29, 30, Gillen, R, *Phys. Rev. B* 97, 165306 (2018), Ovesen, S. et al. *Commun Phys* 2, 23 (2019)] or measurements on homo-bilayers [Hornig, J. et al. *Physical Review B* 97, (2018)., Gerber, I. C. et al. *Physical Review B* 99, (2019)., Sung, J. et al. *Nature Nanotechnology* 15, 750–754 (2020)]. The relatively small values of 10-16 meV we measured may possibly be due to difference in the materials or a smaller interlayer exciton Bohr radius as a result of strong interlayer tunneling.”~~

REVIEWERS' COMMENTS

Reviewer #1 (Remarks to the Author):

The authors have adequately addressed all my technical comments and revised their manuscript accordingly. I support acceptance of this interesting work by Nature Communications.

Reviewer #2 (Remarks to the Author):

The authors have addressed my comments nicely. I believe it is appropriate to publish the current version of manuscript.

Reviewer #3 (Remarks to the Author):

All comments in my previous report have been satisfactorily addressed. I recommend publication of the manuscript in its present form.

"

Reviewer #1 (Remarks to the Author):

The authors have adequately addressed all my technical comments and revised their manuscript accordingly. I support acceptance of this interesting work by Nature Communications.

Reviewer #2 (Remarks to the Author):

The authors have addressed my comments nicely. I believe it is appropriate to publish the current version of manuscript.

Reviewer #3 (Remarks to the Author):

All comments in my previous report have been satisfactorily addressed. I recommend publication of the manuscript in its present form.

We thank all the referees for recommending the publication of the manuscript.